# An Exploring Human Resource Development in Small and Medium Enterprises in Response to Electric Vehicle Industry Development

Chadatan Osatis [ID] and Chonticha Asavanirandorn *

College of Population Studies, Chulalongkorn University, Bangkok 10330, Thailand; chadatan.o@chula.ac.th
*   Correspondence: chonticha.a@chula.ac.th

**Abstract:** Transitioning the automotive industry from internal combustion engines (ICE) to electric vehicles (EV) has arisen as a critical challenge for global countries in achieving human resource development, owing to the need of new labor skills and replacement of automation systems. By applying a case study of Thailand's automotive industry in response to this challenge, we aimed to plot out the trajectory of changes involved in the industry's workforce during its transition with a central focus on Small and Medium-sized Enterprises (SMEs), in order to make sound suggestions to the government on how to build an effective policy for industry growth. An exploratory research design was adopted for the investigation. Both primary and secondary sources were collected. Interviews with key stakeholders, including official agencies, organizations in the business sector, and academic institutions, were conducted in a semi-structured format to collect as primary data. Meanwhile, secondary data were gathered from reports and other scholarly contributions that are relevant. All of the data that were collected were subjected to qualitative methods of analysis, including content and theme analysis. We found that the advances in technology and associated skills have posed challenges to the SMEs for the workforce relocations in terms of occupational shifts and skill development, with engineering demand potentially growing 10% while low labor skills declined by nearly 70%. We emphasized that without effective policies for establishing EV roadmap and coordination practices between public and private stakeholders, this transition would have a detrimental effect on the workforce development of SMEs, which would ultimately have a harmful impact on the automotive industry and the economy.

**Keywords:** human resource development; next-generation automotive industry; skill transition; technological advancement; electric vehicle; roadmap; workforce; Thailand

## 1. Introduction

Development of the electric vehicle (EV) industry is strongly encouraged all over the world in light of the tremendous acceleration of climate change and other environmental problems, especially to reduce carbon emissions. Several initiatives and academic research projects to promote the development have been advocated, and some of them have already begun, e.g., tax incentives and market promotion [1], and development of EV's infrastructure [2,3]. This development will impact the entire automotive supply chain and progress the global automotive industry into the next generation.

Into the next generation, the workforce in the automotive industry needs to acquire new skills in order for it to be in line with the new production process. These include, particularly, high technical skills to support battery management systems, quality assurance, mechatronics, lean automation systems, as well as maintenance of electrical and electronic components. In another perspective, the workforce demand in the existing automotive market will be affected. Considering internal combustion engine (ICE) powertrains, which require approximately 2000 components while EV powertrains require only 20 components [4], this would have an enormous impact on the market and the workforce.

It is essential to investigate workforce relocation in the Small and Medium-sized Enterprises (SMEs) of the automotive industry. Due to the fact that the majority of SMEs serve as supporting producers for the automotive industry's assemblers, EV disruptions in the automotive industry, which require fewer parts and components, will result in a decrease in SMEs' inputs, thereby reducing the demand for labor input in the production process. As a result, it is crucial to address human resource relocation, as certain members of the workforce are required to reskill and upskill in order to remain employed in the industry, while some others who have trouble adjusting are possibly excluded from the industry. Practical solutions are needed for mitigating this difficulty.

At the moment, it is possible to address the fact that there is a lack of research concentrating on this matter, specifically the workforce development of SMEs. In this regard, we aimed to research employment practices and human resource developments in response to the automotive transition. It is anticipated that its contributions would assist policymakers in developing effective policies for facilitating the transition and alleviating the effects on vulnerable workforces.

We applied a case study to Thailand for achieving the research's objective. The main reason is that the automotive and parts industry of Thailand is a significant contributor to its economy, accounting for a large segment of GDP. In 2019, over 2 million cars were manufactured, accounting for 5.8% of GDP [4]. This led to the country being ranked as 11th in world automotive manufacturing and becoming a key producer in the automotive supply chain. In 2016, the ICE automotive companies and its supply chain accounted for 27 assemblers (18 car and 9 motorcycles companies), 710 auto part makers, and 1700 as supporting companies [5]. In total, it supported 890,000 employees in the country [4]. For this reason, Thai SMEs, the majority of which are supporting companies [6], might provide as a useful case study for examining domestic enterprise adaptability.

The rest of the study is divided into six sections. Firstly, the study provides a literature review on factors influencing human resource development. After that, research design and methodology are presented, followed by the results. Then, the study discussion is addressed, followed by suggestions for how Thailand, including other countries, can strengthen human resource development to mitigate the impacts from the disruption, which have been ongoing in the automotive industry.

## 2. Literature Review

Human resource development during a transition period is a significant priority when it comes to upgrading a business. With regard to the transition from ICE to EV production in the automotive and parts industries, a restructure of the labor market is only to be expected, as a significant portion of the ICE workforce would likely have to leave this sector as the transformation is followed by new technologies or manufacturing procedures, decreasing demand for labor input. Workers who stay, on the other hand, would have to enhance their current skills or acquire new skills in order to keep up with the industry's technological advancements.

Taking the latter group into consideration, numerous scholars have established that existing labor's ability to reskill or upskill is driven by their perceptions and attitudes regarding their career choices, capabilities, and anticipated technological advancement [7–11]. As addressed by Fleisher and Wang [9] who examined evidence from China during a transitioning economy, worker motivation to continue their education was substantially bolstered when they began to be paid what they were worth, and so, the less education workers had, the more motivated they were to acquire more (if economically feasible for them). This phenomenon was also noted by Sabirianova Peter [11], who researched a skill-biased transition in the Russian economy in the context of market mechanisms, institutions, and technological change. Findings indicated that the change of the market mechanism from the old socialist wage-setting mechanism to one that adjusts the wage ratio to the actual differences in marginal productivity had a significant effect on the adjustment of skills and labor productivity. The study showed that one way boosting productivity during

the transitory period was to increase wages to reflect market abilities so that the market mechanism of labor demand and supply could enhance the ability to adjust on its own.

In research specific to the automotive industry and its parts, several researchers have proved that skill development must be established as continuous-process education [12–16]. According to [12], who identified the skills needed in the Indian automotive industry, when technical institutions, authorities, and relevant agencies collaborated with industry to develop curricula and establish a common evaluation and certification system after receipt of a diploma, this ensured that diploma holders had acquired the necessary technical and soft-skill capabilities that the market demanded. This conclusion was also supported by [16], who found that businesses, governments, and technical vocational education and training (TVET) institutions needed to create powerful, dynamic, and closely monitored partnerships to enhance and deepen the skills of automotive technology students. The study also advised that a one-year mandatory internship program in relevant industries should be considered for technical students prior to their graduation. This necessity has been confirmed by the importance of training and development programs introduced by [17]. The study highlighted that capacity to change, whether technical or strategic, depends on the competency of the company's members, and training and development may help an organization adapt to these changes more smoothly and rapidly.

More to the point, employment or workplace flexibility has been discussed for industrial and organizational capacity to cope with a change. Flexibility is a topic that has been discussed at length, but in a general sense, it may be described as "the ability of workers to make choices influencing when, where, and for how long they engage in work-related tasks" [18]. The flexibility might be influenced by both employer and employee. The difference between "economic pull" reasons and "social push" is one strategy for introducing greater clarity into the shift towards flexibility. According to [19], flexibility from the employer's perspective refers to the employer's efforts to increase the flexibility of labor utilization in response to cost-related competitiveness concerns. These are called economic pull factors. In the meanwhile, flexibility from the employee's perspective refers to employees' desires for increased flexibility and control over their working life, as well as the need to find a balance between paid and unpaid work. This term refers to social push factors. Both are important as they strongly influence management practices in an organization.

In addition, roles of government interventions are also discussed in the EV development. For automotive industries transitioning to advanced technologies, several studies have reported that government action is an essential mechanism for creating the right environment [4,20–22]. Incisive investigation by [4] explored the electric vehicle trend and its impact on the automotive and parts industries in Thailand. The study recommended that the government should establish a committee to assess the situation and impact of structural changes in the economy, particularly as a result of technological change, and to make policy recommendations that are inclusive of all stakeholders—employees, workers, government, academia, and the general public. This would foster an environment conducive to the automotive sector while it transitions to electric vehicle production. However, to do so, the government needs to take a more active role in assisting with the transition process both in terms of flexible employment practices and human resource development.

## 3. Conceptual Framework

Reflections from the literature survey support that the automotive industry will restructure their production technology in response to the EV development, having massive impact on human resource development. In this sense, we conceptualized our analysis describing in Figure 1.

We hypothesize that the development of EVs would motivate SMEs to restructure their workforces and improve their human resources. This will disrupt the automotive industry in both direct and indirect ways. Direct disruption is directed at ICE production in connection to the decline in demand caused by disrupted automotive parts, whilst indirect disruption is targeted at the changes of demand in reaction to new production or technology

for producing EV parts. In this scope, it would have consequences for massive worker migration in the automotive industry. Therefore, there are two perspectives that should be analyzed in depth: employment flexibility, which represents how SMEs respond to the EV disruption, and transitional impacts on professions and skill, which seek to anticipate proportionate changes in occupations and what skills are required. These may be used as preliminarily empirical data to plan for workforce relocation in response to the change in labor demand.

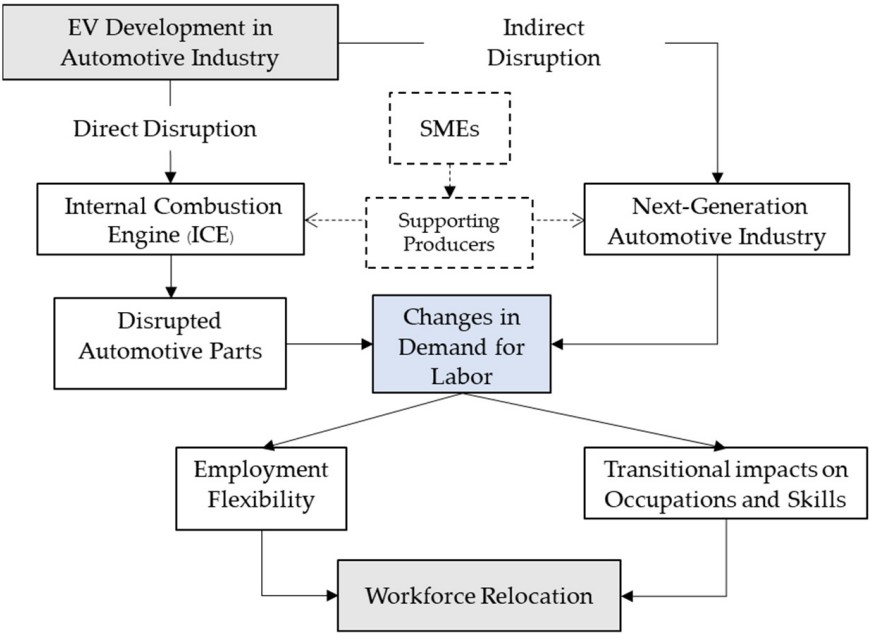

**Figure 1.** Conceptual framework. Source: developed by authors.

## 4. Methods

In order to address emerging challenges facing automotive manufacturing and the industry's response in terms of human resource development, we intend to focus on prevailing conditions facing the industry and on the capabilities needed to adjust employment practices and identify human resource development strategies in a transitioning period. Details of the methodology are as follows.

### 4.1. Participants

We recruited 15 key informants: 8 executive board of directors/top management officers from Small and Medium Enterprises in automotive industry, 4 experienced officials from automotive associations, 1 senior government official, and 2 representatives from automotive training institutions. The aim here was to investigate existing employment practices and skills development challenges faced by the industry and to come up with an adaptation of employment practices and a path for human resource development that can best serve the next generation of the industry.

### 4.2. Data Collection and Validation

To achieve our objectives, data were employed from both primary and secondary sources. Primary data were collected from semi-structured interviews during July–December 2021. The topics for the interview were forwarded to the key informants. The main topics included "What are factors and challenges influencing your human resource management?", "How does your business adapt in response to EV disruption?", "How do government interventions support your human resource development", and "How would you suggest practical recommendations to support SMEs' human resource development?".

Before using the information, the interview reports were returned to the informants for checking of accuracy. This was to ensure that the data collected were correct and that the key informants accepted this. In addition, informants were also asked for permission to disclose their identities, in terms of their positions or perspectives. Additionally, reflections from secondary data sources such as academic research and news and press releases were gathered as supplementary sources of analysis.

After extracting data from the interview, the study assessed the data's reliability employing Denzin [23], the Triangulation approach, which establishes the convergence of data from different sources. Two forms of triangulation were employed in this study: data triangulation and methodological triangulation. The aim of data triangulation is to ensure the accuracy of information obtained from several sources while accounting for differences in time, place, and participant inputs. Methodological triangulation is used to demonstrate that data acquired using distinct approaches are accurate. The study analyzed data using interviews and observations, and then documented and analyzed the informant's responses and perspectives.

At the last verified process, the triangulation results were provided to the informants for reading or re-asking (in some cases) to ensure that the data were reliable and consistent, and sufficient to assess and summarize for the study's findings.

### 4.3. Analytical Frame

This study intended to primarily explore existing conditions of human resource development faced by the automotive industry in a period of transition from ICE to EV, applying a case study of Thailand. This attempts to suggest the practical strategies for the government, implementing a policy towards the upcoming disruption in automotive industry effectively. This study, therefore, followed the concept of exploratory research design, which is an effort to explore critical challenges arising from existing conditions that influence employment practices and human resource development of the industry. Hence, data to be analyzed were qualitative in nature.

In terms of analysis, thematic analysis and content analysis techniques with theoretical frameworks were adopted. Thematic analysis was utilized in order to investigate the transitory consequences on occupations and skills experienced by SMEs. Since data are limited due to competition concerns, and especially when information about electric vehicle adaptation is private and proprietary, statistical analysis is constrained. Consequently, performing research based on themes is the most effective method for conveying both the current impacts and their developments.

In terms of contents analysis, flexibility contents were adopted. Although flexibility in employment in labor markets can be investigated by various techniques depending on the context of the study, in this study, we adopted the flexibility analysis proposed by [24]. The technique consists of four components of employment flexibility. First is numerical flexibility to adjust the amount of labor in correspondence with fluctuations in demand and technological change. This flexibility discerns business making decisions on number of employees in the company. Second is functional flexibility to transfer jobs easily and incentivize workers to multi-task according to fluctuations and changes in demand. This flexibility is defined as the ability of employees to execute tasks or jobs in a timely manner. The next is wage flexibility, that is, to adjust wages with ease, corresponding to changes in the labor market and competitive circumstances. Last is temporal flexibility, i.e., to employ workers with ease under different kinds of labor contract and being able to adjust the number of workers in line with fluctuations in demand. The four types of flexibility noted above represent important factors in employers' decisions on the flexibility of employment practices and the pathway of human resource development in response to changes that occur during the transitioning period.

### 4.4. Procedures

Flowchart of the conducted research is described in Figure 2, indicating four steps to achieve the objective of the study.

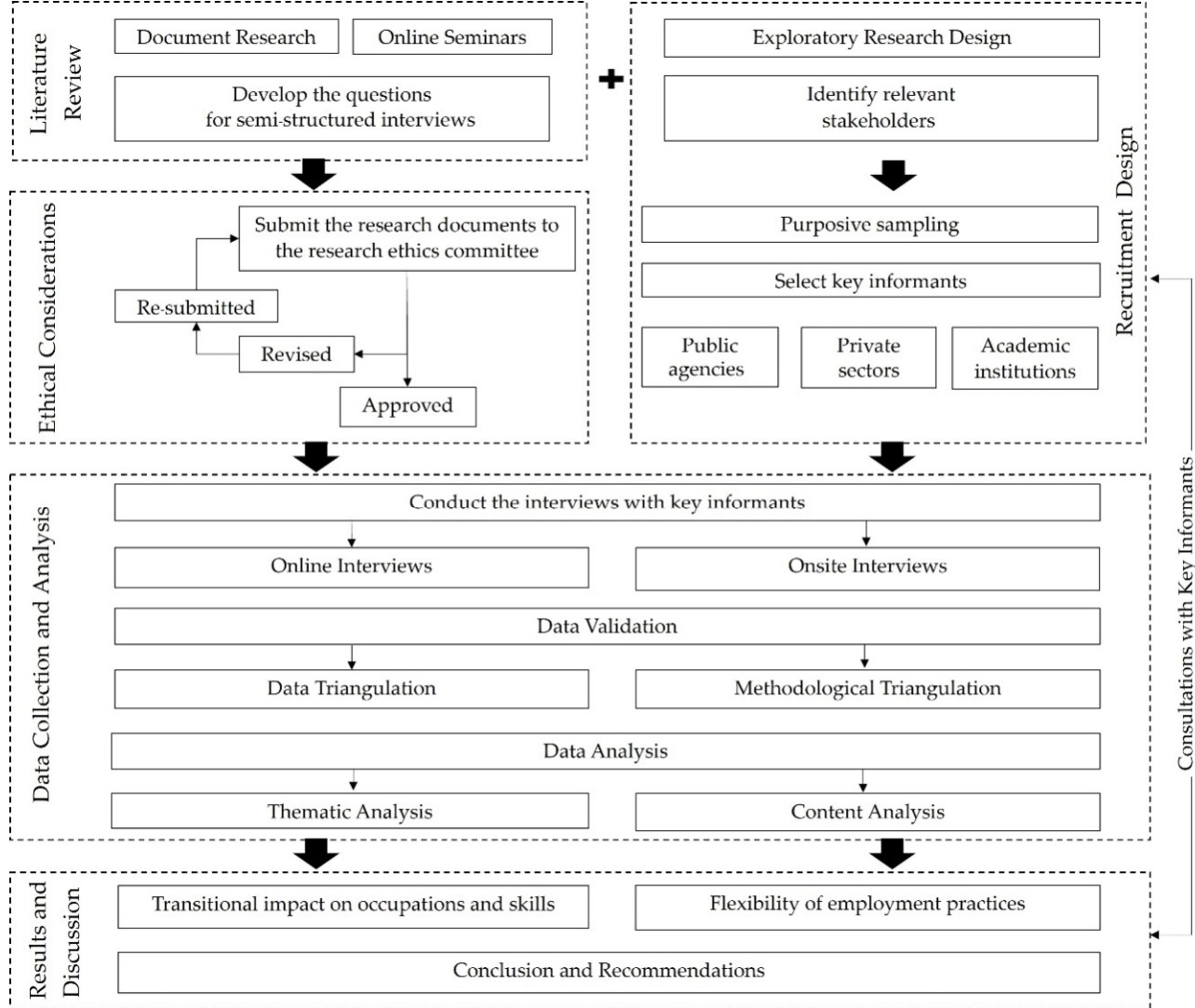

**Figure 2.** Flowchart of the conducted research. Source: summarized by authors.

We began by researching relevant studies and participating potential online seminars in order to develop the questions for semi-structured interviews as well as to cope with recent progress on EV development. In the meantime, we identified key informants by a purposive sampling technique based on their experiences, role or position, and capabilities. They were selected from three segments including public agencies, private sectors, and academic institutions.

To ensure that conducting this research was in accordance with ethical practices, developed questions for the semi-structured interviews were forwarded to the respective ethical committees for their considerations. After its approval, we conducted the interviews with key informants both online and onsite. Data validation was performed through data triangulation and methodological triangulation as explained in Section 4.2.

At the step of data analysis, the content analysis method was used for the results and discussions on flexibility of employment practices; the thematic analysis method was used for transitional impact on occupations and skills among the workforce in the industry. The final step was to derive the conclusion and recommendations on effective

measures to support human resource development in the industry in its transitioning period. Consultations with key informants were organized in order to guarantee that key findings were practical, and reflect on the context of SMEs' response.

## 5. Results

The study collected, verified, and analyzed information according to the analytical frame. Sections 5.1 and 5.3 were generated by thematic analysis, whereas Section 5.2 was investigated by contents analysis. Key findings are presented in the following sections.

### 5.1. The Transitional Impact on Occupations and Skills

Due to the fact that Thailand established an agenda to promote electric vehicles by promoting them for general public use since 2013 and targeting 2035 as the year by which all new vehicles, including both those used for public transport as well as private cars and trucks, will be EVs [25], and preparing for low-carbon society in accordance with the European Union's Carbon Border Adjustment (EU's CBAM) [26], various national policies have been implemented, with the target of reaching a hundred percent of new EV vehicle registration by 2035. Development of workforce capability to support EV production has already been initiated. Most challenges in promoting skills to achieve inclusive growth relates to the limited scope of the existing skills policy, which focuses heavily on the supply side of skills [27]. As the industry begins to cope with changes in technological know-how, the success of human resources development will be influenced by existing occupations and skills. Table 1 presents the four groups predicted to be affected and that need to reskill and upskill.

The first affected group is highly skilled engineers, comprising those working in quality assurance, quality management, and various fields such as design, storage, and energy, and also product design. Transitioning to new skills is quite advanced and requires technology transfer from global carmakers. In terms of products and parts, new skills involve transitioning to working with hybrid engines, lightweight bodies and suspension, or battery management systems. In terms of process, engineers must work with a lean automation system integrator, artificial intelligence, or mechatronic programming and coding. Employment in this area represents an increase from 10 to 20 percent of the workforce in the industry. However, there is a challenging issue concerning a shortage of labor in these fields resulting from a limited pool of professional trainers. In addition, some training in these skills has run into intellectual rights proscriptions owing to the proprietary rights and fierce competitiveness in the industry.

The second and third groups are technicians. Here, the employment demand reflects an upward trend from 20 to 50 percent and requires a multi-skilled workforce able to use advanced technological tools and know-how. Occupations at the skilled non-manual technician level include jobs in logistics and supply chain management, warehousing, etc. Such workers are expected to have skill types associated with programmable logic control (PLC), enterprise resource planning (ERP), material requirement planning (MRP), and supply chain management (SCM). Nevertheless, skill development at this level depends on an individual's productivity and attitude towards skill acquisition. As a result, the efficacy of skill transfer should be supported by all relevant stakeholders. As for skilled manual technicians, demand for this group has declined and requires an effective upskilling of the existing workforce. Occupations here include supervisors, production workers, maintenance workers, parts assemblers, and machine operators. The required skill types mainly include basic IT, digital skill, and relevant soft skills, for instance mind management, industrial discipline, safety, problem-solving, and communication.

**Table 1.** Transitional impact on occupations and skills by each skill level.

| | Skill Levels | Occupations | Major Skill Types | Major Challenging Issues | Preliminary Change in Labor Demand |
|---|---|---|---|---|---|
| Engineer | Highly skilled | • Quality assurance, quality management representative • Engineer: design, storage and energy • Product designer | • Hybrid engine, lightweight body, suspension, motor brake, battery management system • Lean automation system integrator, artificial intelligence, mechatronic, programming and coding | Sharply increasing labor demand but shortage of labor supply | Increase from 10% to 20% |
| Technician | Skilled non-manual | • Logistics and supply chain management • Warehouse • Technicians • Information technology (IT) | • PLC, ERP, MRP, SCM * • Sensor, relay, timer, and pneumatic | Multi-skilled workforce which is compatible with new technology | Increase from 20% to 50% |
| | Skilled manual | • Supervisors • Production • Maintenance • Parts assemblers • Machine operators | • Basic IT or digital skills • Multi-skills tasking • Soft skills tasking | Effective upskilling of existing workforce | |
| Operator | Low skilled | • Labors in the production line • Quality control | • Basic IT or digital skills • Multi-tasking skills | Decreasing employment trend and issues related to workforce relocation | Reduce from 70% to 30% |

Source: Preliminary results based on fieldwork interviews. Note: * PLC (Programmable Logic Control), MRP (Material Requirement Planning), ERP (Enterprise Resource Planning), and SCM (Supply Chain Management).

Finally, low-skilled operators are noted as a vulnerable workforce as they are most at risk of being replaced by automation and robotics. Demand for this group will sharply decrease from 70 to 30 percent. In the transition period, they might be forced to leave the industry due to difficulty in adapting to technological advancement. These occupations are related to labor-intensive positions and most involve performing repetitive tasks such as those associated with plant and machine operation.

*5.2. The Flexibility of Employment Practices*

According to our investigation on prevailing conditions pertaining to transitioning expectations as presented in Table 1, it is noticeable that many EV and EV-related firms have begun reorganizing their work patterns. We have observed, in terms of the flexibilities of employment practices, the follows:

• Numerical flexibility

We observed that some firms have decided to start imposing numerical flexibility on low-skill labor by adopting partial automation. The total number of workforces on a particular production will be in line with how effective the automation system implemented is. As automation systems require specialists, numerical flexibility places a significantly positive impact on highly skilled engineers and skilled non-manual technicians. In contrast, a negative impact is placed on low-skilled operators and somewhat on skilled manual technicians who do not work directly with the automated system.

- Functional flexibility

According to the interviews, firms have provided a comprehensive training course for engineers and technicians to facilitate the upgrading of their skills to meet a requirement of multiple functions. This makes labor more productive and able to multi-task in response to integrative solution requirements and changes in demand. For example, an engineer who is responsible for digital manufacturing and IoTs program has to integrate the related know-how, including lean IoT systems and factory IoT data management. Likewise, inspector technicians have to broaden their skills related to lean manufacturing with measurement processes and also digital and vision measurement technology in accordance with an integrative solution.

- Wage flexibility

Certain firms have adjusted wages according to competitive circumstances to ensure that workers who participate in a reskilling or upskilling program will receive appropriate remuneration for their new abilities. Wage flexibility thus addresses and adheres to the notion of skill recognition through certification, testing, and a qualification system rather than the traditional Thai compensation and benefit structures based on formal educational achievement over actual, practical capabilities. This motivates labor to eagerly participate in training activities.

- Temporal flexibility

We found that temporal flexibility has been handled through labor contracts to increase more flexibility and reliance on subcontracting, especially for non-core tasks. This is foreseen as administrative strategy to help employers reduce costs and improve productivity. Workers who cannot keep up with technical skills in relation to (EV) production may be permitted/asked to retire early in order to boost a firm's performance. In compliance with existing labor laws, severance payment is provided. Assistance in recruiting is probably offered.

Although the rise of SME's flexibility kept increasing in response to the EV disruption, it has been observed that most Thai SMEs in the automotive industry have strongly faced a limitation on adopting wage and numerical flexibilities. In terms of wage flexibility, its limitation is contributed by labor law. As a matter of fact, SMEs are rigid to labor contracts and minimum wage rates guaranteed by the Labour Protection Act B.E.2541 (as amended), and it is difficult for SMEs to adjust wage flexibly. In the meantime, contributed by organizational structure, this situation indirectly stipulates implicit cost arising from restructuring the workforce in the organization, especially cost related to upskill and reskill training. As a result of this, changing the number of employees is rigid to be flexible in SMEs.

*5.3. The Pathway of Human Resource Development*

The pathway for human resource development in EV production is shown in Figure 3. Taking into consideration when existing workers in ICE production demand to upskill or reskill to meet the EV production technology, this challenge potentially forces some incapable labor to relocate either to continue working in the companies in different positions or to leave from the industry due to difficulties in upgrading or adding the new necessary skills.

In the meantime, with our considerations moving forward for high-capacity labor or new entrants for the EV development, we were informed that there is a government initiative, a five-year action plan (2021–2025) on the development of human resources to support the next-generation automotive industry under the cooperation agreement between Ministry of Industry, Ministry of Higher Education, Science, Research and Innovation, and Office of the Vocational Education Commission, according to Deputy Director of the Office of Industrial Economics, Ministry of Industry, to support this transition. This initiative encouraged both the private sectors and academic institutions to seek collaboration. Equipping high-capacity ICE workers by providing in-house training with specific

courses for assisting reskilled and upskilled process, includes, for example, introducing employment careers or internship programmes to potential vocational school students.

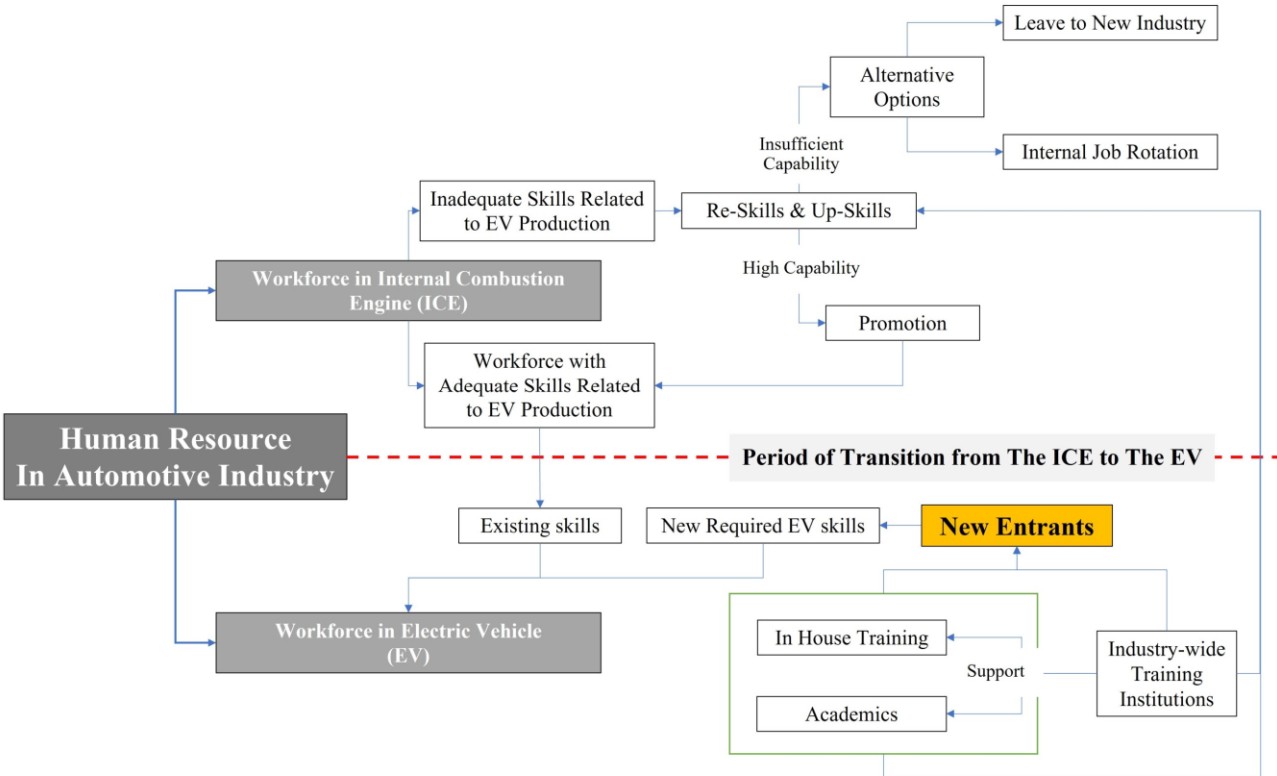

**Figure 3.** The pathway of human resource development in EV production. Source: developed by authors according to fieldwork interviews.

However, with regard to government policy, we found that especially those in top management and from leading organizations are not satisfied with the country's roadmap for human resource development owing to the unclear direction. Although Thailand has initiated a national EV policy committee to draw up a policy roadmap to help the country achieve 100% zero-emission vehicle production by 2035, at the moment, there is only policymaking; however, without a clear roadmap, enterprises are unsure how to prepare for the future. As a result, human resource development obtained just cursory planning. In addition, this includes a lack of governmental incentives and knowledge management strategies. As a result, developing human resources appropriate for EV production continue to be a rough uphill struggle. Although Thailand currently promotes knowledge exchange between Thai entrepreneurs and foreign entrepreneurs, the initial problem of Intellectual Property remains a major obstacle to human resource development. In response to this, the associations and respective agencies have started to cooperate with EV research agencies/companies to develop knowledge related to EV parts by themselves, such as the development of electric battery properties. However, they are likely to gain few contributions resulting from this collaboration.

## 6. Discussion

The study explored the pathway of SMEs' human resource development in response to the progress of EV market. As observed, SMEs in the automotive supply chain are recognized as a supporting producer, with some of them being contract manufacturers, which means that the direction of its development correlates with demand from the main actors of the industry, i.e., automotive assemblers/producers.

It is possible to address that many SMEs have delayed their human resource development. One reason might be as a result of roles of SMEs in the supply chain. As a supporting producer, it has caused the SMEs not to independently grow their human resource development unless there is a clear development from their automotive assembler. Especially where specific skills to support the EV manufacturing have to be acquired from abroad, it is highly challenging for them to be trained in response to the EV development. As a result of this, with a case study in Thailand where the major assemblers are foreign companies, the country currently has faced a shortage of EV manpower due to lack of knowledge transfer from the automotive leaders. It is expected to have a massive impact on supply shortage for engineering, the demand of which is expected to dramatically increase 10%–20% in the near future. In the worst case, some SMEs might be left from the automotive industry since they do not have sufficient human capital to support development of the industry, especially when those SMEs are performing tasks relating to engines, transmissions, exhaust systems, fuel systems, alternators, and starters, for example.

Moreover, it has been noticed that the flexibilities of SMEs in response to the automotive industry development have been based on technological advancements in tandem with the growth of EV market. Rapid progress on automation systems might be a key contributor. Thai SMEs that produced components for the ICE and are still making them in EV show that these enterprises still intend to restructure their employment practices, such as replacing workers with automation systems, restructuring wage systems, or shifting to outsourcing employment. This confirms that the automotive industry's employment will be reformed sooner or later, with much impact to SMEs as they are dependent on their assemblers. In consequence, a huge number of workers will be reduced, and some former automotive industries might be forced to lay off due to being unable to adapt to new technology. Those workforces are identified as vulnerable people, particularly as for those who are in aging or pre-aging periods. This infers that training programmes must be prepared for those vulnerable people, the aging workforce in particular, in order to support them for their early retirement or to have a job when leaving the automotive industry.

Furthermore, a roadmap for EV development is important for the industry's next stage of development, specifically positioning the country on whether to continue expanding EV production or maintaining ICE manufacturing. According to a case study from Thailand, SMEs in the industry have struggled with human resource development due to an unclear roadmap. In other words, a clear roadmap would assist companies in developing the capability to adapt to modern technology. In the host country of vehicle assemblers similar to Thailand, a working group comprised of members' familiar with the whole industry supply chain must be formed in order to produce a policy that more clearly defines the direction, particularly the role of the country as a key component in the chain.

## 7. Conclusions and Recommendations

Meeting the demands of the burgeoning electric vehicle industry, its production requirements, employment practices, and human resource development has emerged as one of the country's most problematic concerns. Applying a case study from SMEs' human resource development in Thailand, its contributions are addressed toward the global countries to take effective measures to support the industry in its transitioning period. Most SMEs have adopted flexibilities of employment practices in line with the impacts on workforce relocation and also required skills for the changes. Recommendations are highlighted as follows:

1. The blueprint for change and the roadmap for EV development have to be clear and in line with the capacity of the existing ICE automotive industry. The roadmap has to include:

    a. Taskforce committees need to be set up for determining the direction of human resource development and supporting existing entrepreneurs and workers as they transition.

b. The taskforce should be inclusive and engage all relevant stakeholders, including government authorities, private sectors, academic institutions, and employee representatives—bearing in mind that nominated participants have an obligation to put aside personal conflicts of interest.

c. Incentivize creation of strategies for human resource development such as corporate tax reduction, co-investment in skill development, and co-designing for training.

d. A fund for knowledge management needs to be established—for providing professional training to relevant stakeholders, recruiting expatriates from abroad for knowledge sharing on technological advancement, etc.

e. A training database system should be established to encourage individuals and firms to access training resources to catch up with industrial trends and lessen in-house training costs.

2. At the industry level, inter-industry as well as industry-academia strategic partnerships with various kind of collaborations need to be established, particularly the collaboration on sharing mechanical, software, electronics and electrical instruments. This is in order to reinvent and customize research know-how to make it more relevant to marketable production. The investment of carrying out R&D is relatively infeasible for SMEs. To facilitate it, public R&D and testing centers where the SME entrepreneurs can access machines and other resources need to be established subsequently. All this could boost resource sharing and help EV-related innovation and a knowledge ecosystem flourish.

3. At the organizational level, training that is in line with the characteristics of the industry should be facilitated. In small and medium enterprises (SMEs), development strategies should comprise comprehensive training suited for a variety of needs. Given that most SMEs lack funds for developing their human resources, partial support is not very effective in assisting them. Particularly, incentive strategies for human resource development and a fund for knowledge management should be the first priority concerns.

Though electric vehicle manufacturing has begun around the world, including in the country we are focusing on, Thailand, it has not yet reached its full potential. There is still a need for scholars across the world to initiate research that further investigates the ways in which business and other relevant public agencies can assist in increasing employment opportunities for workers who are being forced to leave the automotive industry. Such deeper understanding can help them increase their chances of being employed in other industries. In addition, further research can investigate the development of E-bike usage on business transformation, especially issues related to workforce relocation. This is in light of the fact that the market for bikes would be disrupted by electric vehicles, which is similar to the automotive industry.

**Author Contributions:** Conceptualization, C.O.; data collection, C.A.; verifying result, C.A.; drawing the conclusion and recommendations, C.O. All authors have read and agreed to the published version of the manuscript.

**Funding:** This Research is funded by Thailand Science Research and Innovation Fund (TSRI Fund) (CU_FRB640001_01_51_1) and the Ratchadapisek Sompoch Endowment Fund (763008) from the Collaborating Centre for Labour Research, Chulalongkorn University (CU-ColLaR).

**Institutional Review Board Statement:** This research adheres to strict ethical standards. The participation in the survey is voluntary. Interviewees' information is kept confidential. All interviewee data and questionnaires are anonymized and maintained securely in accordance with the ethical certification no. 222/2564 from The Research Ethics Review Committee for Research Involving Human Research Participants Group 1, Chulalongkorn University.

**Data Availability Statement:** Not applicable.

**Acknowledgments:** The research team would like to pay our gratitude towards CU_TSRI Fund, the key interviewees and all relevant stakeholders who kindly supported for this research project.

**Conflicts of Interest:** The authors declare no conflict of interest.

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
