# Peer review of "An Exploring Human Resource Development in Small and Medium Enterprises in Response to Electric Vehicle Industry Development"

_wevj, doi:10.3390/wevj13060098_

Round 1

Reviewer 1 Report

This is an interesting manuscript, whose eventual publication on this journal I will be going to support with this statement. Cover:

  1. The transition towards electric vehicles has some implications in terms of reduction of carbon emissions that have not been covered at all in the document. A policy discussion about this can include the following literature: Cavallaro F., Danielis R., Nocera S., Rotaris L. (2018). Should BEVs be subsidized or taxed? A European perspective based on the economic value of CO2 emissions. Transportation Research Part D: Transport and Environment 64: 70-89; Foley, A., Tyther, B., Calnan, P., Ó Gallachóir, B. (2013). Impacts of Electric Vehicle charging under electricity market operations. Applied Energy 101, pp. 93-102; Samaras, C., Meisterling, K. (2008). Life cycle assessment of greenhouse gas emissions from plug-in hybrid vehicles: Implications for policy. Environmental Science and Technology 42(9), pp. 3170-3176;
  2. There is also room to include e-bikes in the narrative of this paper, as they ultimately belong to the Electric Vehicle Industry. This is a point that has gained some interest in the electric scientific narrative: Bruzzone, F., Scorrano, M., Nocera, S. (2021). The combination of e-bike-sharing and demand-responsive transport systems in rural areas: A case study of Velenje. Research in Transportation Business & Management 40, 100570; Fishman, E., Cherry, C. (2016). E-bikes in the Mainstream: Reviewing a Decade of Research. Transport Reviews 36(1), pp. 72-91; Winslott Hiselius, L., Svensson, Å. (2017). E-bike use in Sweden – CO2 effects due to modal change and municipal promotion strategies . Journal of Cleaner Production 141, pp. 818-824;

3 I feel like the argument should be generalized outside of Thailand to gain the interest of the readers of this journal. Would it be possible to add some words in the conclusions?

Author Response

Response to Reviewer 1 Comments

This is an interesting manuscript, whose eventual publication on this journal I will be going to support with this statement. Cover:

Thank you very much for your supportive comments and kind recommendations. Major revised contents and information have been included in our paper. In addition to that, we would like to give some detail of our revision as follows:

Point 1: The transition towards electric vehicles has some implications in terms of reduction of carbon emissions that have not been covered at all in the document. A policy discussion about this can include the following literature: Cavallaro F., Danielis R., Nocera S., Rotaris L. (2018). Should BEVs be subsidized or taxed? A European perspective based on the economic value of CO2 emissions. Transportation Research Part D: Transport and Environment 64: 70-89; Foley, A., Tyther, B., Calnan, P., Ó Gallachóir, B. (2013). Impacts of Electric Vehicle charging under electricity market operations. Applied Energy 101, pp. 93-102; Samaras, C., Meisterling, K. (2008). Life cycle assessment of greenhouse gas emissions from plug-in hybrid vehicles: Implications for policy. Environmental Science and Technology 42(9), pp. 3170-3176;

Response 1: We have updated our literature review, incorporating your advice indicated in lines 34-36, and added some relevant literature according to suggestions in the references (lines 529-533).

Point 2: There is also room to include e-bikes in the narrative of this paper, as they ultimately belong to the Electric Vehicle Industry. This is a point that has gained some interest in the electric scientific narrative: Bruzzone, F., Scorrano, M., Nocera, S. (2021). The combination of e-bike-sharing and demand-responsive transport systems in rural areas: A case study of Velenje. Research in Transportation Business & Management 40, 100570; Fishman, E., Cherry, C. (2016). E-bikes in the Mainstream: Reviewing a Decade of Research. Transport Reviews 36(1), pp. 72-91; Winslott Hiselius, L., Svensson, Å. (2017). E-bike use in Sweden – CO2 effects due to modal change and municipal promotion strategies. Journal of Cleaner Production 141, pp. 818-824;

Response 2: Your advised literature has broadened our knowledge in the field of study. It is an interesting issue that may be further explored; therefore, we included the E-bike issue in the direction of future research. (Line 510-513)

Point 3: I feel like the argument should be generalized outside of Thailand to gain the interest of the readers of this journal. Would it be possible to add some words in the conclusions?

Response 3: Thanks to your suggestion, we have improved the sentence and amended key points to gain more interest from the reader, highlighting in lines 459-461, and also we have amended a rationale of Thailand’s case in the introduction in lines 63-72.

We feel grateful for your kind and supportive comments and suggestions. All the suggestions from the reviewers have broadened our knowledge, giving us the opportunity to learn to improve our manuscript and the hope for publishing our work in the Special Issue "Feature Papers in World Electric Vehicle Journal in 2022". 

Thank you very much.

Faithfully Yours. 

Reviewer 2 Report

The authors of the article presented research on employee development in enterprises producing electric vehicles. This is an interesting issue because, for ecological reasons, this industry began to develop dynamically. Certainly, this development will be evan more dynamic in the coming years.

The authors analyzed the changes faced by employees in small and medium-sized enterprises. For this purpose, research methods such as interviews, questionnaires, and observations were used.

 After reading the article, I have the following comments:

  • The article lacks a detailed description of the methods used. Authors should separate a Methods section and describe the tools used, e.g. the type of questionnaires used.
  • Literature should also be completed.
  • The authors do not even summarize the questionnaires that were used. It is not known what questions were asked, and in what form the results were obtained.
  • The statistical analysis of the obtained results has not been presented.
  • The obtained results were not validated.
  • The authors write about the growing demand for employees with specific abilities. How was this information obtained, to what period does it relate?
  • Conclusions should be supplemented with specific results obtained during the analysis.

Before publication, the article must be supplemented, because it is missing essential content.

Author Response

Response to Reviewer 2 Comments

The authors of the article presented research on employee development in enterprises producing electric vehicles. This is an interesting issue because, for ecological reasons, this industry began to develop dynamically. Certainly, this development will be even more dynamic in the coming years.

The authors analyzed the changes faced by employees in small and medium-sized enterprises. For this purpose, research methods such as interviews, questionnaires, and observations were used.

 After reading the article, I have the following comments:

Thank you very much for your deliberate suggestions. We have updated our revision and would like to clarify as follows:

Point 1: The article lacks a detailed description of the methods used. Authors should separate a Methods section and describe the tools used, e.g. the type of questionnaires used.

Response 1: Our methodology section is improved and provides more details. Major revisions are as follows:

  • We added a conceptual framework in Section 3 (lines 144-165). This is in order to provide our hypothesis for the investigation.
  • In addition, to make a reader more understandable our investigation, we have added the research procedure (indicated in section 4.4, lines 242-265)

Moreover, the method section is rearranged into 4 parts comprising participants, data collection and validation, analytical frame, and procedures in order to deliberately described the details of our method (lines 166-265). Additionally, we clearly identified the research tool (semi-structured interview) and the questions we used to ask the key informants (lines 183-189).

Point 2: Literature should also be completed.

Response 2: A literature review has been developed. We have included the literature in relation to the employment flexibility, indicated in lines 119-132.

Point 3: The authors do not even summarize the questionnaires that were used. It is not known what questions were asked, and in what form the results were obtained.

Response 3: According to your deliberate advice, we have rechecked our manuscript and found our errors since we identified the questionnaire as one of our research tools, together with interview and observation (Line 183-185). In fact, the questionnaire as a tool for research is not included in our study. All data used are obtained from the interviews and observations, and results are analyzed by content and thematic analysis techniques (Line 196-204 and Line 218-219). Therefore, we revised section 4 of Method and clearly clarified all the content mentioned above and we added the topics of questions in accordance with your suggestions. (185-189)

Point 4: The statistical analysis of the obtained results has not been presented.

Response 4: Our study adopted the exploratory research design in order to explore the pathway of EV disruption in the automotive industry. Therefore, most of the analysis is qualitative by nature. However, taking this concern into consideration, we adopted the triangulation method for very statistical information (indicated in section 4.2, Line 182-207)

Point 5: The obtained results were not validated.

Response 5: The results and key findings have been forwarded to our key informants to verify whether it represents the SME context of the automotive industry or not. We have updated this process in section 4.4  and lines 262-265.

Point 6: The authors write about the growing demand for employees with specific abilities. How was this information obtained, and to what period does it relate?

Response 6: Preliminary results in Table 1 are obtained by the interview with the selected key informants and we also confirmed the result through consultation with the key informants. The period of data collection is specified in section 4.2 which is July - December 2021). Line  184-185, 285

Point 7: Conclusions should be supplemented with specific results obtained during the analysis.

Response 7: Our specific results are discussed in section 6 (discussion) lines 411-455. Therefore, our conclusion is derived from the discussion in order to make a recommendation.

Point 8: Before publication, the article must be supplemented, because it is missing essential content.

Response 8: Thanks to your deliberate comments, we have amended the essential contents of our paper, i.e. conceptual framework (section 3), the procedure of conducting research (section 4.4), key reflections from the result, in terms of theoretical implications on flexibility in the context of Thailand (section 5.2; lines 361–370),  and generalized a conclusion for a reader (line 459-461). Thank you very much for your valuable suggestions to help improve our manuscript.

We feel grateful for your kind and supportive comments and suggestions. All the suggestions from the reviewers have broadened our knowledge, giving us the opportunity to learn to improve our manuscript and the hope for publishing our work in the Special Issue "Feature Papers in World Electric Vehicle Journal in 2022". 

Thank you very much.

Faithfully Yours. 

Reviewer 3 Report

Dear Authors, this paper is very well written. I like the paper and thank you for the opportunity to read yours work.

This article is of extremely high scholarly standards, it is original in numerous respects and certainly most timely in terms of World Electric Vehicle Journal relevance. Key area of improvements is editorial and contextual. By point of view:

  • editorial - using less complex sentence structures and clarifying specialist terms would make the article more 'reader friendly- an ensure its appreciation amongst wider readerships.
  • issues for further work - the conclusions should  draw the essence of the study from the propositions made and recommend further direction for future research directions.

My recommendation is to be accepted for publication after minor revision. 

Best regards!

Author Response

Response to Reviewer 3 Comments

Dear Authors, this paper is very well written. I like the paper and thank you for the opportunity to read yours work.

This article is of extremely high scholarly standards, it is original in numerous respects and certainly most timely in terms of World Electric Vehicle Journal relevance. Key area of improvements is editorial and contextual. By point of view:

Point 1: editorial - using less complex sentence structures and clarifying specialist terms would make the article more 'reader friendly- an ensure its appreciation amongst wider readerships.

Response 1: Thank you very much for your supportive comments and impactful suggestions. We are aware of “reader friendly” thanks to your kind suggestion. We have generalized some specific terms/words/sentences to be general and more simple in order for assisting readers to better understand. However, further English proof to help edit the revised manuscript would be more helpful for the readers. And we have already taken this into our consideration.

Point 2: issues for further work - the conclusions should draw the essence of the study from the propositions made and recommend further direction for future research directions.

Response 2: Thanks to your advice, we have drawn the essence of our study proposition and clearly identified the sentences in the conclusion (Line 461-463). In addition, we added further direction of future research. (Line 510-513)

My recommendation is to be accepted for publication after minor revision.

Best regards!

We feel grateful for your kind and supportive comments and suggestions. All the suggestions from the reviewers have broadened our knowledge, giving us the opportunity to learn to improve our manuscript and the hope for publishing our work in the Special Issue "Feature Papers in World Electric Vehicle Journal in 2022". 

Thank you very much.

Faithfully Yours. 

Reviewer 4 Report

The paper is dealing with exploring HRD in SMEs in the electric vehicle industry. In my opinion, paper should be further improved in order to be acceptable. Therefore, please incorporate the following reqs.:

-The abstract is loosely written and it should be rewritten. Without leaving any doubt, a standard abstract must present the paper's objective precisely; the source of data (which is not present in your abstract), the analytical approach used; key findings, and any policy implications and recommendations. At this moment this is not the case. 

-Research gap in the Introduction, as well as research motivation, should be better explained and in more detail. 

-Literature review is not carried out well and should be improved by incorporating relevant and up-to-date references. 

-Flowchart of the conducted research should be introduced and it would be beneficial for the readers. This should be differ from Fig. 1.

-Conclusions are fine, but directions for future research should be presented. Also, what are the limitations and theoretical implications?

Author Response

Response to Reviewer 4 Comments

The paper is dealing with exploring HRD in SMEs in the electric vehicle industry. In my opinion, the paper should be further improved in order to be acceptable. Therefore, please incorporate the following reqs.:

Thank you for your kind advice. We have updated our paper as follows:

Point 1: The abstract is loosely written, and it should be rewritten. Without leaving any doubt, a standard abstract must present the paper's objective precisely; the source of data (which is not present in your abstract), the analytical approach used; key findings, and any policy implications and recommendations. At this moment this is not the case.

Response 1: For the structure of the standard abstract, We have rewritten the abstract in accordance with the sequence and elements you advised; research objective, data collection, analytical frame, key finding, and recommendation (Line 12-27).

Point 2: The research gap in the Introduction, as well as research motivation, should be better explained and in more detail.

Response 2: This research intends to plot the trajectory of human resource development in SMEs in response to the EV disruption in the automotive industry in Thailand. This is to bridge the gap in human resource development for the next generation of the automotive industry. We identified the research gap and explained our research motivation in more detail in the  Introduction (Line 48-62)

Point 3: The literature review is not carried out well and should be improved by incorporating relevant and up-to-date references.

Response 3: We have amended and incorporated relevant and up-to-date references in the section of the literature review (line 79) and references (line 528) according to your advice.

Point 4: A flowchart of the conducted research should be introduced, and it would be beneficial for the readers. This should be different from Fig. 1.

Response 4: According to your impactful and valuable advice, we inserted the flowchart of the conducted research in Section 4.4 and explained the step of the research procedure (Fig.2 and Line 242-265). Thank you very much.

Point 5: Conclusions are fine, but directions for future research should be presented. Also, what are the limitations and theoretical implications?

Response 5: We have revised to identify the direction of future research (Line 505-513) and clarified the limitations and theoretical implications of our study’s results in 5.2 (Line 361-370). We are grateful for your kind suggestions to help us revise a better version of our manuscript.  

We feel grateful for your kind and supportive comments and suggestions. All the suggestions from the reviewers have broadened our knowledge, giving us the opportunity to learn to improve our manuscript and the hope for publishing our work in the Special Issue "Feature Papers in World Electric Vehicle Journal in 2022". 

Thank you very much.

Faithfully Yours. 

Round 2

Reviewer 1 Report

Paper improved following my recommendations. Ready for publication

Reviewer 2 Report

Accept in present form.

Reviewer 4 Report

Thank you for improving your manuscript by incorporating my suggestions.